# Exploring Therapeutic Advances: A Comprehensive Review of Intestinal Microbiota Modulators

**DOI:** 10.3390/antibiotics13080720

**Published:** 2024-08-01

**Authors:** Lara Pires, Ana M. González-Paramás, Sandrina A. Heleno, Ricardo C. Calhelha

**Affiliations:** 1Centro de Investigação de Montanha (CIMO), Instituto Politécnico de Bragança, Campus de Santa Apolónia, 5300-253 Bragança, Portugal; laravaqueiro@ipb.pt (L.P.); calhelha@ipb.pt (R.C.C.); 2Laboratório Associado para Sustentabilidade e Tecnologia em Regiões de Montanha (SusTEC), Instituto Politécnico de Bragança, Campus de Santa Apolónia, 5300-253 Bragança, Portugal; 3Grupo de Investigación en Polifenoles, Área de Nutrición y Bromatología, Facultad de Farmacia, Universidad de Salamanca, Campus Miguel de Unamuno s/n, 37007 Salamanca, Spain; paramas@usal.es

**Keywords:** dysbiosis, therapeutic interventions, probiotics and prebiotics, new microbiome-modifying, gut modulation

## Abstract

The gut microbiota establishes a mutually beneficial relationship with the host starting from birth, impacting diverse metabolic and immunological processes. Dysbiosis, characterized by an imbalance of microorganisms, is linked to numerous medical conditions, including gastrointestinal disorders, cardiovascular diseases, and autoimmune disorders. This imbalance promotes the proliferation of toxin-producing bacteria, disrupts the host’s equilibrium, and initiates inflammation. Genetic factors, dietary choices, and drug use can modify the gut microbiota. However, there is optimism. Several therapeutic approaches, such as probiotics, prebiotics, synbiotics, postbiotics, microbe-derived products, and microbial substrates, aim to alter the microbiome. This review thoroughly explores the therapeutic potential of these microbiota modulators, analysing recent studies to evaluate their efficacy and limitations. It underscores the promise of microbiota-based therapies for treating dysbiosis-related conditions. This article aims to ensure practitioners feel well-informed and up to date on the most influential methods in this evolving field by providing a comprehensive review of current research.

## 1. Introduction

The human microbiota is a constantly changing ecosystem that forms after birth. It consists of microorganisms that inhabit the body’s surfaces and cavities, engaging in mutually beneficial interaction with the host [1]. The intestinal microbiota, which contains trillions of microorganisms, primarily anaerobic bacteria, exhibits the greatest diversity of microorganisms in the human body [2]. This ecosystem is mainly composed of *Firmicutes*, *Bacteroidetes*, *Actinobacteria*, and *Proteobacteria*, with *Bacteroidetes* and *Firmicutes* constituting more than 90% of its composition [3]. The composition of this ecosystem undergoes continuous changes throughout life. Right after birth, the baby’s digestive system starts to be populated with microorganisms, a process influenced by the mode of delivery and whether the baby is breastfed [4]. By the age of three, the intestinal microbiota attains an adult-like composition, with unique variations among individuals, yet maintaining relative stability during adulthood [5] (Figure 1). While each person possesses a distinct microbial species composition, the overall phylogenetic profile is categorised into various host–microbial ecosystems with functional disparities, impacting responses to diet and medications [6].

The microbiota profoundly impacts host health by fortifying a competitive barrier against pathogens, metabolising nutrients, modulating immune responses, and participating in neural communication [6]. Primarily metabolising carbohydrates into short-chain fatty acids (SCFAs), the microbiota significantly impacts energy harvest and lipid digestion, synthesising essential vitamins and regulating bile acid metabolism [9]. Furthermore, it regulates immune homeostasis by modulating innate and adaptive immunity and neurological function through the interaction of gut–brain axes, such as anxiety, depression, cognitive disorders, and autism spectrum disorders (ASD) [10,11].

Dysbiosis is a disorder of the digestive tract and bacterial flora concerning the endobiogenic needs of the body [12]. Dysbiosis can be caused by insufficiency or loss of diversity in commensal flora or pathogenic flora competing with the commensal microbiome [13]; various factors, including diet, antibiotics, and ageing, can trigger that. This imbalance has the potential to result in a range of ailments, such as diabetes, neurological disorders, and metabolic disorders. [14]. Acquiring a thorough understanding of dysbiosis is essential as it underscores the need to maintain gut physiological microbiota homeostasis, prevent diseases, and promote general well-being. The beneficial effects are usually accomplished by the action of many gut modulators [15]. Modulating intestinal microbiota is complex. Probiotic species and prebiotic substrates have considerable potential in preventing many diseases and enhancing overall well-being. Prebiotics and probiotics continue to receive significant attention in scientific research, healthcare practices, and the public consciousness [16].

For instance, clinical studies have shown promising results for prebiotics like inulin and fructooligosaccharides and probiotics such as *Lactobacillus* and *Bifidobacterium*. The probiotic sector is expected to develop at a substantial rate of 7% per year, while the prebiotic industry is projected to increase by 12.7% over the next eight years [17]. These data indicate that both therapies are projected to see significant expansion in the future. The statistics, acquired from the latest market research, emphasise the growing popularity and potential of prebiotics and probiotics, driven by an expanding repertoire of candidate species and substrates tailored to target newly elucidated microbial niches and host interactions [18].

Amidst the growing popularity of probiotics and prebiotics, it is crucial to retain an objective perspective. While bioactive chemicals are used in many consumer products, including food, beverages, and topical therapies, the scientific validation of their health advantages differs. There still needs to be a more substantial improvement in the general public’s understanding of the terminology and mechanisms of action. This emphasises the need to exercise prudence and use analytical reasoning when confronted with enticing but unsupported claims about health [19]. To address these challenges, the International Scientific Association for Probiotics and Prebiotics (ISAPP, https://isappscience.org/, accessed on 11 June 2024), a non-profit organisation, has successfully implemented standardised terminology and promoted scientific discourse that brings together scientists, healthcare professionals, and industry representatives in expert panels to promote the understanding and appropriate use of probiotics and prebiotics via publications and conferences [20].

Additionally, the field of microbiomes extends beyond probiotics and prebiotics, including various innovative medicines. A new era of “biotics” biogenics is emerging, and it can revolutionise healthcare practices (Figure 2 and Table 1). Increasing symbiotic formulations, which synergistically mix prebiotics and probiotics to amplify their benefits, provides an exciting way to improve microbial therapeutics’ efficiency [21].

Recently, there has been a growing interest in postbiotics, which the ISAPP describes as “a product made from non-living microorganisms and their parts that provide a health benefit to the host”. The area of research focuses on exploring the therapeutic benefits of gut metabolites [24]. Postbiotics, including antimicrobial peptides, vitamins and short-chain fatty acids (SCFAs), demonstrate rapid health benefits. For example, butyrate, a derivative of SCFA, has anti-inflammatory properties and benefits the gastrointestinal system’s homeostasis [25].

New methods such as microbial consortia [26], communities of microorganisms that work together, live biotherapeutic products, and genetically modified organisms (GMOs) are increasing the options for changing the microbiome [27]. Microbial consortia may be engineered to exhibit certain functionalities, such as enzymatically degrading particular dietary substrates or synthesising advantageous chemicals. Live biotherapeutic products (LBP) contain viable microorganisms that can cure or prevent illnesses [28]. Genetically modified organisms (GMOs) are creatures that have been genetically altered in a way that does not occur usually. These technologies provide customised approaches to target specific health issues and unique requirements [26].

When used with breakthroughs in personalised nutrition and precision medicine, these therapies can transform healthcare delivery by changing health and illness microbial indicators. Given these recent advancements, it is increasingly important to guarantee the excellence and honesty of probiotic and prebiotic products [29]. This has led to the development of innovative quality assurance methods to accurately assess these products’ dosage, viability, and effectiveness [30]. These procedures include thorough testing to detect particular strains, verifying that the goods contain the stated quantity of live microorganisms, and evaluating the viability and performance of these microorganisms. The demand for these treatments is increasing rapidly in many products, ranging from foods to medications. As a result, searching for effective ways to modify the microbiome is an active area of scientific research and innovation.

This review aims to analyse new research results to clarify the bioactive effects, underlying processes, and therapeutic implications of probiotics, prebiotics, and developing microbiome modulators. Although there is an effort to examine these influential treatments comprehensively, it is essential to acknowledge that the field of microbiome research is moving quickly, with continual advancements and discoveries. As a result, this assessment might only cover some of the latest developments up until 2023. Nonetheless, it is expected that this evaluation will contribute to improving the understanding of the effectiveness of these treatments in shaping the future of healthcare and enhancing human health.

## 2. Probiotics and Prebiotics: From Concepts to Chronology—Entering the New “Biotic” Era

The mechanisms by which probiotics and prebiotics operate are intricate, diverse, and often specific to the strain or compound used. The International Scientific Association of Probiotics and Prebiotics (ISAPP) defined probiotics in 2014 as “living microorganisms that, when given in sufficient quantities, provide a positive effect on the health of the host” [31]. This definition slightly amended the one previously proposed by the FAO/WHO in 2001 [23] (Figure 3). Probiotics commonly used include *Bifidobacteria, Lactobacilli* and other lactic-acid-producing bacteria (LAB), which are primarily derived from fermented dairy products and the human gut microbiome. Fermented products included in the human diet like kefir and yoghurt provide various probiotic strains. Probiotics encompass a wide array of bacteria, including *Lactobacillus*, *Lactococcus*, *Leuconostoc*, *Pediococcus*, *Propionibacterium*, *Bifidobacterium*, *Bacillus*, certain *Streptococcus* and *Enterococcus* species, as well as the *Saccharomyces* yeast [32]. These bacteria and yeast are commonly found in nature and are considered potential sources of LAB probiotic strains. Consuming probiotics has been linked to significant health benefits, including reduced risk of obesity, cardiovascular diseases, and type 2 diabetes, offering promising prospects for future healthcare [33].

Probiotics engage with both the host and the microbiome through molecular effectors found on the cell structures or released as metabolic products. Probiotic metabolites can influence the microbiota by engaging in cross-feeding interactions, modifying the gastrointestinal microenvironment, competing for nutrients and binding sites, and inhibiting growth through the production of specific antibacterial compounds like bacteriocins [34]. Regarding host cells, probiotic effector molecules can directly interact with receptors on immune cells, intestinal epithelial cells, vagal afferent fibres and enteroendocrine cells. These interactions result in specific effects within the gut, such as improving the integrity of the intestinal barrier and modulating inflammation. Furthermore, probiotics exert broader effects on the body by involving the immune, endocrine, and nervous systems [35]. Additionally, probiotics can also participate in the enzymatic breakdown of substances produced by the host, such as bile salts and ingested foreign compounds. Probiotic bacteria also possess distinct compounds on their surface, such as lipoteichoic acids, exopolysaccharides, and surface-layer proteins, which contribute to their efficacy. Several of these compounds are specific to certain bacterial strains and are responsible for producing effects unique to those strains [36].

According to the recommendations established by the World Health Organisation (WHO), probiotic foods should include at least 10^6^ cells per gram or millilitre of product when consumed. Moreover, the recommended therapeutic dosage is 10^8^–10^9^ cells/g of the product [37]. It is essential to mention that the microorganisms inside must be able to withstand the effects of gastric juice and bile salts. Once probiotics have crossed this chemical barrier, they should attach themselves to the intestine’s surface to carry out their beneficial functions for promoting health [36]. Probiotic products stimulate the nonspecific cellular immune response by activating natural killer cells and macrophages, which release different cytokines. In addition, they can enhance the gut mucosal immune system by increasing the population of IgA (+) cells [38]. Additionally, probiotics can also facilitate digestion and lactose hydrolysis, enhance mineral assimilation, and promote the biosynthesis of various vitamins such as thiamine, riboflavin, niacin, pantothenic acid, and vitamin K. These functions demonstrate their antiproliferative, proapoptotic, and antioxidative properties [39]. These significant health benefits of probiotics and prebiotics reassure us about their potential healthcare applications, instilling confidence in their effectiveness.

The United States Food and Drug Administration (FDA) has classified probiotics as Generally Regarded as Safe (GRAS) at the strain level. Conversely, the European Food Safety Authority (EFSA) has categorised them as species-specific under the Qualified Presumption of Safety (QPS) [40]; however, this classification does not apply to new probiotic species that have not been previously used. Submitting through GRAS, QPS, and novel food frameworks may pave the way for commercialisation. Moreover, novel regulatory frameworks are emerging for pharmaceutical applications. In particular, the FDA is currently classifying live biotherapeutic products, and the European Directorate for the Quality of Medicines is also participating in this process [41].

Advancements in affordable complete genome sequencing and robust cultivation methods have enabled the isolation and characterisation of a new variety of microorganisms from human microbiomes. These microorganisms can provide health benefits and could be developed as next-generation probiotics or pharmabiotics [42]. Pharmabiotics are bacterial cells originating from humans or their products that have been proven to have a pharmacological effect on health or disease. Various bacteria isolated from the human gut, including *Faecalibacterium prausnitzii*, *Bacteroides* spp., *Roseburia intestinalis*, *Akkermansia muciniphila*, and *Eubacterium* spp., have shown promise for their probiotic potential. These bacteria constitute a significant proportion of the currently cultivable human gut microbiome and offer physiological functions not always directly provided by bifidobacteria or lactobacilli. The conversion of these species into commercially viable probiotics poses challenges due to their need for nutrient-rich growth media and anaerobic environments [43]. Nevertheless, *A. muciniphila* stands out as a promising option already available in combination with inulin [44], *Bifidobacterium longum* subsp., and other anaerobic bacteria. This combination has been shown to improve glucose levels in individuals with type 2 diabetes. Researchers are investigating other potential interventions to restore microbial populations, maintain physiological homeostasis in disease states, and explore the gut microbiome as a source of new candidate probiotic strains. Fermented foods may serve as a reservoir of probiotics due to their abundance of naturally occurring lactic acid bacteria (LAB) strains [45]. The LAB strains offer notable health advantages, including the mitigation of metabolic syndrome (obesity), type 2 diabetes (TDM2), and cardiovascular illnesses (CVD). Potential future probiotics may be derived from a wide range of sources, including fruits, vegetables, grains/cereals, dairy, meat and fish products, and honey, and environmental factors such as soil. The field of microbiome research is constantly evolving and holds the ability to usher in a new era of living organisms. This has generated enthusiasm and curiosity about the future of scientific investigation [46].

Introduced by Gibson and Roberfroid in 1995 [47], prebiotics originated from the observation that certain carbohydrates, selectively fermented in the colon, can promote the growth of lactobacilli and bifidobacteria which are known to have beneficial effects on health. Prebiotics are food ingredients or substances that cannot be digested and benefit living organisms by explicitly facilitating the development and activity of specific bacterial species. The classical prebiotic effects occur when specific groups within the microbiota consume the substrate, promoting their growth and metabolic activity. The supply of nutrients to specific groups of bacteria can also indirectly impact other bacterial groups in the microbiome. Consequently, it can stimulate growth through interactions involving the sharing of resources and hinder the growth of pathogens by displacing them [48]. Administering prebiotics results in changes in the microbial composition and metabolites, impacting several host signalling pathways, including those in the endocrine, epithelial, immunological and neurological systems. The use of prebiotics may provide several health benefits, such as enhancing intestinal function, boosting immune response, regulating glucose and lipid metabolism, promoting bone health, and controlling hunger [49].

Omics methods have advanced in recent years, enhancing the ability to study the effects of prebiotics in both laboratory and clinical settings, including in vitro and in vivo research as well as human clinical trials [50]. Prebiotics are now targeting a more comprehensive range of microbial responders, including candidate health-promoting genera such as *Akkermansia* spp., *Christensensella* spp., *Roseburia* spp., *Faecalibacterium* spp., *Propionibacterium* spp., and *Eubacterium* spp., expanding beyond LAB. These prebiotics can directly or indirectly promote the growth of these and other bacterial groups through cross-feeding interactions.

Currently, a narrow range of confirmed prebiotic substances exists, with galactans and fructans dominating the market [51]. The desire to stimulate a wider group of commensal organisms has led to the development of novel candidate prebiotic compounds, which likely include carbohydrate-based substances derived from plants, those that mimic animal-derived substrates, yeast-based substances, and many non-carbohydrate substances, including polyphenolics, fatty acids, herbs, and other micronutrients [52].

Polyphenols and other intestinal microbiome-modulating carbohydrates, such as resistant starch, polydextrose, xylo-oligosaccharides, pectin, and human milk oligosaccharides (HMOs), are considered prebiotic candidates but have not yet met the ISAPP consensus definition [53]. Over 8000 known polyphenols exist in plants, vegetables, and fruits, and many reach the colon intact to be utilised by resident microorganisms. Some polyphenols show prebiotic potential, such as cinnamon and cranberry extracts, which stimulate A. muciniphila or provide antimicrobial action against pathogens [54].

Prebiotics will likely be isolated from novel sources as a focus on sustainability, cost, and scale emerges [55]. The cumulative annual production of 1.3 billion tonnes of food waste in the food chain presents a valuable and environmentally friendly reservoir of natural bioactive components. Many side streams from fruit, vegetable, and grain processing contain potential prebiotics [56]. Future prebiotic compounds may undergo chemical or structural modification through sonication, high pressure, acid treatment, enzyme treatment, and oxidation to alter their functionality. As of June 2024, there were 497 clinical trials (ClinicalTrials.gov) for prebiotics and 2147 registered clinical trials with probiotics. The number of studies and investigational targets suggests significant investment in developing prebiotics and probiotics as bioactive ingredients or supplements for a range of potential applications [17].

Several papers have proposed alternative definitions of prebiotics, with a broader scope to better integrate emerging microbiome-modulating compounds [57]. In this case, however, blurring the lines between prebiotics and fermentable fibres would undermine the original concept of prebiotics, which is to selectively boost the growth of certain microorganisms while also providing health benefits. Other nonfermentable modulators of the microbiome also lie adjacent to the prebiotic scope and will play a role in the future of microbiome modulation [58].

While vitamins are usually absorbed in the small intestine, their administration in large amounts or through colon-targeted formulations can modulate the colonic microbiome, as has been demonstrated with both riboflavin and niacin [59]. Furthermore, genomic studies have suggested that B vitamin exchange may be a component of normal symbiotic relationships among gut microbial species, and several lines of evidence suggest a potential corrective role for colonic vitamin administration in some disease states [60].

Synbiotics and complex mixtures, combining the effects of fermentable substrates and live microorganisms, are known as synbiotics. These are composed of a combination of an accepted prebiotic and probiotic, where their mechanisms of action can be independent of each other, and both the prebiotic and the probiotic must have their own demonstrated health benefits [61]. Synergistic synbiotics contain a fermentable substrate for the co-administered live microbe, where the substrate and the microbe may or may not be able to elicit a health benefit independently of each other. Synbiotics hold promising therapeutic potential in various research areas, including obesity, cardiovascular diseases, gastrointestinal disorders, and cancer [62]. Research has demonstrated that concurrently using probiotics and prebiotics can diminish inflammation, enhance gastrointestinal symptoms, and decelerate the progression of chronic kidney disease (CKD) by regulating the gut microbiome. Furthermore, studies have demonstrated that synbiotics can decrease uremic toxins and biochemical markers linked to CKD, leading to improved clinical outcomes [63]. The efficacy of synbiotics depends on the specific characteristics of the selected probiotics and prebiotics and their synergistic interactions in the intestinal environment. Common synbiotic combinations include bacteria from the genera Bifidobacteria or Lactobacilli paired with prebiotics like fructooligosaccharides (FOS), which are known for stimulating the growth of beneficial strains [64]. Synbiotic therapy has been compared to a multiple-drug regimen for the gut, offering a potent means of combating diseases mediated by dysbiosis. By combining live microorganisms with microbial “fertilisers,” synbiotics can effectively target and modulate the gut microbiota, promoting overall health and well-being.

The development of novel strains and substrates will influence the future of synbiotics, targeting vacant microbiome niches in individuals and subgroups. These advances have potential applications in both gastrointestinal and ex-gut sites, similar to the probiotic and prebiotic fields. The microbiome can be thought of as naturally occurring mixtures of fermented foods, which provide microbes, microbial substrates, and a variety of bioactive fermentation metabolites [65]. The increasing popularity of such foods will likely drive everyday consumer recognition of prebiotics, probiotics, and synbiotics moving forward.

The critical area of microbiome research that looks for individual and group microbiome signatures to predict disease incidence, progression [66], and response to treatment has been made possible by the efficient and robust analysis of large amounts of data. Distinct taxonomic profiles, with specific genera and species, have been linked to health and disease states, host biomarkers, dietary habits, and lifestyle characteristics. According to these data, there is a notable interest in using specific methods to control the microbial makeup within individuals or particular groups of people. Probiotics and prebiotics are promising interventions that can redirect these signatures towards better health [67] using various modes of action. Above all, probiotics have a minimal effect on the microbiome composition in healthy individuals. However, new probiotics derived from microbiomes could be a way to change the adult microbiome in the future. The best candidates so far are health-related microbes identified using top-down methods [68]. It is crucial to understand that probiotics can provide health benefits independently of microbiome colonisation or modulation, and their ability to affect the microbiome is not essential for their utility. Prebiotics can also aid in correcting compositional imbalances by encouraging the growth of under-represented species. While traditional prebiotics are mainly known for their bifidogenic effects, cross-feeding interactions have shown that inulin can also affect other specific intestinal microbiome groups, such as *Faecalibacterium* spp. and *Anaerostipes* spp. [69].

In the future, novel and emerging prebiotic compounds can be used in targeted ways to manipulate the microbiome and its metabolic outputs. Many of the structural characteristics of prebiotics are known to influence which microbes can utilise the substrate, including monosaccharide structure, degree of polymerisation, branching, linkages, and addition of functional groups or other modifications [70]. To aggregate data from disparate studies, Lam and Cheung [71] proposed creating a multidimensional prebiotic structure–microbiome matrix, sequentially testing and collating data from prebiotic interventions and mapping the resulting microbiome impact from structural variation. This information could be teamed with machine learning to predict the structural characteristics of a prebiotic required for the modulation of specific microbiome profiles, leading to custom prebiotic and synbiotic production based on these characteristics. Data collection and predictive modelling could also capture microbial metabolic interactions, layering in the potentially complex ecosystem effects of designer prebiotic administration and mixtures.

Taxonomic microbiome characterisation is increasingly combined with [72] metagenomic or metabolomic data to understand the functions microbes might perform [72]. Integrated data sets may assist in identifying the loss of microbiome functions, or vacant “functional niches”, essential to host health, providing further potential for precision medicine intervention as shown by the new biotic era [73].

The advent of the “biotics” era has introduced a myriad of terms and concepts in the field of microbiome research (Table 1), reflecting the dynamic and evolving nature of this science. Among these, modulators stand out, prompting scientists to create subdivisions and corresponding new terms based on the targeted illness or system.

Postbiotics, which comprise non-viable microbial cells, cellular compounds, and soluble or metabolic byproducts derived from probiotics, offer health benefits without the stringent manufacturing or storage conditions required for live microorganisms [74]. This makes them particularly suitable for incorporation into food products, especially in resource-limited regions.

The identification of postbiotics has led to numerous terms in the scientific literature, such as non-viable probiotics, non-biotics, proteobiotics, pharmabiotics, metabiotics, parapsychobiotics, paraprobiotics, inactivated probiotics, and ghost probiotics [75]. These terms highlight the diverse nature of postbiotic compounds and their potential applications in the food and pharmaceutical industries. Postbiotics can be classified based on their origin and chemical composition into categories including molecules derived from probiotic cells, such as peptidoglycan-derived muropeptides, exopolysaccharides (EPS), teichoic acids, surface protruding molecules, secreted proteins/peptides, bacteriocins, cell-free supernatants, organic acids, vitamins, short-chain fatty acids (SCFAs), neurotransmitters, and biosurfactants. Each postbiotic has established chemical structures and offers unique health advantages, exerting local effects on specific intestinal epithelial tissues and impacting multiple organs and tissues [76].

The therapeutic potential of postbiotics lies in their ability to replicate the beneficial effects of probiotics while mitigating the risks associated with live microorganisms, especially in individuals with compromised immune defences or altered intestinal barriers. Furthermore, postbiotics offer better stability and shelf life than live probiotics [77].

Further research can progress this field by improving the characterisation of biological reactions to probiotics and prebiotics in clinical trials, thus expanding our understanding of these interventions and their potential for precise application. There is a demand for greater utilisation of integrated, multi-omic methods to analyse and understand the effects of probiotics and prebiotics, including metagenomic, metatranscriptomic, and metabolomic techniques.

## 3. Exploring Modulatory Agents of the Gut Microbiota: Current Insights

Recently, the scientific community has shown increased interest in the unique and important function that the gut microbiota plays in both maintaining good health and contributing to illness. The limitations of conventional medicine in treating disorders linked to the gut microbiota have generated an urgent need for new therapeutic approaches. Progress in metagenomic, metatranscriptomic, and metabolomic approaches contributes to the development of effective medicines with a significant focus on treating dysbiosis [78].

Gut modulators (probiotics, prebiotics, synbiotics, and the emerging “new biotics”), aim to restore gut microbiota, showing a great potential in treating several diseases that impact the immune, endocrine, digestive, cardiovascular, and nervous systems. Furthermore, their impact on the gut–brain axis suggests potential uses in the treatment of neuropsychiatric illnesses, including anxiety, depression, and Alzheimer’s disease [79].

Probiotics, prebiotics, postbiotics, and synbiotics, which come from the microorganisms in the human gut or from food, are considered safe for long-term usage. Extensive research and statistical data indicate that natural medicines, such as gut modulators, often exhibit reduced drug interactions and toxicity compared to synthetic pharmaceuticals [25]. This makes them an acceptable option for patients who are on multiple medications (polymedication) and for the elderly. Unlike conventional drugs that may lead to tolerance, side effects, or toxicity, these substances are more appropriate for long-term usage and pose a lower risk for vulnerable populations, such as children, pregnant women, and the elderly. Furthermore, gut modulators have a favourable pharmacokinetic profile as they are not significantly altered by stomach pH and maintain stability under various environmental circumstances, including temperature [80].

Clinical research has thoroughly investigated the impacts of gut modulators, with a particular focus on their underlying processes. Nevertheless, the categorisation and use of dietary supplements as medicine have been restricted due to differing legislation, management standards, and degrees of public acceptability in different nations. This has resulted in limitations on their research and clinical usage [81].

Multiple studies highlight the therapeutic capacity of these modulators. Based on animal studies, scientists have found that probiotics like *Bifidobacterium longum* and *Limosilactobacillus fermentum* may change the gut microbiota and improve the intestinal mucosal barrier [82].

Despite the significant therapeutic potential of probiotics, prebiotics, synbiotics, and postbiotics, their clinical use is hindered by many challenges. Strain specificity, lack of standardisation, quality control difficulties, and regulatory issues provide significant impediments. Moreover, customised treatment and extensive investigation are necessary in light of cultural and religious perspectives, safety considerations, potential pharmacological interactions, and individual differences. Overcoming these obstacles is crucial to fully maximise the therapeutic benefits of these medications in clinical settings [83].

Ultimately, exploring substances that can modify the gut microbiota, such as probiotics, prebiotics, synbiotics, and postbiotics, offers a hopeful path for treating diseases and improving overall health. To enhance understanding across various research studies, Table 2 provides evidence from specific clinical trials recorded in the ClinicalTrials.gov database. This research includes those that have yielded the most promising results in the last decade. The search was conducted using academic platforms such as PubMed, Scopus, Google Scholar, and ScienceDirect, focusing on studies from 2018 to the present that have garnered significant consensus within the International Scientific Association for Probiotics and Prebiotics (ISAPP). Their capacity to rebalance gut microbiota and address diverse illnesses showcases their promise as future pharmaceuticals. Further investigation, progress in regulatory measures, and thorough clinical testing will be essential in leveraging the advantages of these innovations and overcoming current obstacles, eventually revolutionising our methodology for controlling and treating diseases.

The information in Table 2 about probiotics highlights significant findings regarding their potential to treat various health conditions. Probiotics can aid post-surgical recovery and weight management, highlighting their role in metabolic health. Various probiotic strains also show benefits in reducing body fat, improving lipid profiles, and enhancing insulin sensitivity, indicating that targeted probiotic therapies could be developed for metabolic conditions such as obesity and diabetes, providing a novel approach to metabolic health management [92,93,94].

Within the framework of the gut–brain axis, which refers to the intricate communication system between the gut and the brain, probiotics like *Bifidobacterium* and *Lactobacillus* strains have shown the ability to enhance cognitive functions and alleviate symptoms of neuropsychiatric disorders. It underscores the potential of probiotics in treating mental health disorders through gut–brain axis modulation, offering a complementary approach to traditional psychiatric treatments. Moreover, certain probiotics improve gut barrier function and modulate immune responses in IBS and IBD, demonstrating that probiotics can effectively manage inflammatory bowel conditions, potentially reducing the need for long-term use of anti-inflammatory drugs [107,108].

The role of probiotics in the gut–heart axis is also notable. Modulating gut microbiota may help manage hypertension and reduce cardiovascular disease risk, highlighting the systemic impact of gut health on cardiovascular conditions and suggesting that probiotics could be integrated into cardiovascular disease prevention strategies [39,110]. In addition, probiotics can potentially lessen the severity of liver disease [114] and improve cancer treatment through immune modulation and microbiota balance. In managing chronic kidney disease, probiotics may help manage serum uric acid levels and oxidative stress, indicating a potential role for probiotics in managing this condition and offering a complementary approach to conventional therapies [116].

Various prebiotics, such as inulin and chicory oligofructose, show benefits in enhancing beneficial gut bacteria and reducing obesity-related parameters [123]. These data support using prebiotics in weight management programs, potentially offering a natural and safe way to combat obesity. In the context of diabetes, resistant dextrin and other prebiotics improve glucose metabolism and insulin sensitivity, highlighting prebiotics as a promising intervention for diabetes management and offering an alternative or adjunct to pharmacotherapy. Prebiotics like whole garlic help modulate lipid profiles and reduce dyslipidaemia [132], a condition characterised by abnormal levels of lipids in the blood. This suggests they could play a role in cardiovascular health by enhancing cholesterol metabolism and reducing the risk of atherosclerosis. Regarding the gut–brain axis, xylooligosaccharides and fructooligosaccharides improve cognitive function and gut microbiota composition [136], indicating that prebiotics can significantly impact mental health, with dietary interventions to enhance cognitive functions and reduce neuroinflammation.

Finally, prebiotics such as inulin and glycolipids from tilapia heads show benefits in managing hypertension and reducing markers of inflammation, reinforcing the link between gut and cardiovascular health [139]. This suggests that prebiotics could be beneficial in managing hypertension and reducing cardiovascular risk. While probiotics and prebiotics hold promise for therapeutic benefits, it is crucial to weigh these against potential negative impacts on the gut microbiome and overall health. Probiotics, in particular, can upset the delicate balance of the gut microbiota. The introduction of new bacterial strains can potentially outcompete or suppress native microorganisms, leading to dysbiosis. This imbalance can trigger adverse gastrointestinal symptoms and may even counteract the intended therapeutic effects.

It is important to note that while generally beneficial, probiotic supplementation can carry potential risks. Some individuals may experience adverse reactions, such as bloating, gas, and abdominal discomfort. These reactions, though usually mild and transient, can sometimes worsen the symptoms of conditions like IBS. Probiotics can pose a risk of infection for individuals with insufficient immune systems or underlying health conditions [156]. Although these cases are rare, it is essential to be mindful of the possibility of problems such as bacteremia and fungemia, especially with strains like *Lactobacillus* and *Saccharomyces*. This awareness can help ensure the safe use of probiotics, especially in vulnerable populations. Probiotics can interact with the host’s metabolism in unpredictable ways. While they may enhance certain metabolic functions, there is a risk that they could also interfere with nutrient absorption or alter metabolic pathways, leading to unintended health consequences [157]. While probiotics are often used to modulate the immune system positively, there is a potential for adverse immune responses. Some strains may provoke an exaggerated immune reaction, contributing to inflammation rather than mitigating it. This is particularly relevant in conditions where the immune system is not functioning as it should, a state known as immune dysregulation. In such conditions, probiotics may not have the intended positive effects on the immune system [158].

In conclusion, the tables collectively illustrate the broad therapeutic potential of probiotics and prebiotics across various health conditions. The promising results from clinical trials underscore the importance of further research and development in this field. To fully realise the clinical benefits of these gut modulators, we must address challenges such as strain specificity, standardisation, and regulatory hurdles. However, the potential for further research and development in this field is vast, and future studies should prioritise large-scale clinical trials, personalised medicine approaches, and overcoming regulatory barriers to integrate natural therapeutics into mainstream healthcare. Continued research and clinical trials are required to comprehensively comprehend the advantages and potential dangers of probiotics and prebiotics in therapeutic applications to guarantee their safe and efficient utilisation in clinical practice.

## 4. Future Perspectives: Tackling Global Healthcare Challenges with Probiotics and Prebiotics

Given the increasing healthcare difficulties we currently encounter, there has been a notable focus on the potential efficacy of probiotics, prebiotics, and the new “biotics” as therapeutic interventions. While these substances offer potential health benefits, it is important to consider the risk and limitations associated with any pharmabiotics [156]. Although these medications are generally stable and unaffected by factors such as temperature or stomach acid pH, for example, probiotics may not be suitable for persons with weakened immune systems due to their potential to promote infections in immunocompromised patients. Prebiotics, when used in excess, may lead to gastrointestinal discomfort, emphasising the need for careful use [157].

Assuming the evolving properties of microbiological threats and the rising incidence of antibiotic resistance, it is necessary to prioritise innovative approaches such as the latest generation of ‘biotics’, which have demonstrated greater safety and precision in their therapeutic effects [158]. Antimicrobial resistance (AMR) presents worldwide healthcare problems, negatively impacting the therapeutic value of current medications and making medical treatment harder to achieve. Probiotics and prebiotics can mitigate antimicrobial resistance (AMR) by reducing the need for antibiotics and limiting the spread of antibiotic-resistant bacteria [159]. This is achieved by maintaining a healthy balance of gut flora and boosting immune function. Additionally, the antimicrobial substances produced by probiotics and postbiotics offer novel methods for handling antimicrobial resistance (AMR) [160].

The recent global pandemic has highlighted the importance of proactive measures in preventing and managing illnesses [161]. Probiotics and prebiotics are known to strengthen the immune system and mitigate the severity of viral infections; however, their capacities extend beyond this. Recent research suggests that some probiotic strains possess the capacity to provide defence against respiratory infections and reduce the severity of inflammatory reactions. Besides this, probiotics and prebiotics demonstrate promising results in managing mental health conditions such as anxiety and depression, offering an alternative strategy for immunisation tactics and instilling optimism for enhanced mental health results [162].

The study of probiotics, prebiotics, and biogenics has been significantly affected by improvements in research techniques. Advanced sequencing methods such as metagenomics and transcriptomics enable an extensive survey of the gut microbiota and its effects on the host’s physiology. Artificial intelligence (AI) and machine learning algorithms (ML) accelerate data analysis and prediction formulation, enhancing the capacity to identify novel probiotic strains and pharmaceutical targets [163]. Modern agricultural tools, including microfluidic platforms and organoid models, provide an essential understanding of the complexities of microorganisms and the symbiotic relationship between hosts and diseases.

Ensuring product quality is crucial for maximising healthcare efficiency. Setting standards and benchmarks for probiotic and prebiotic products is a significant task undertaken by regulatory agencies such as the FDA, WHO, and ISPP. Their crucial role in this sector is substantial and should be recognised [164].

By enhancing consumer confidence and streamlining market access, their efforts ensure that only safe and effective products can enter the market. Quality assurance includes the process of describing strains, verifying compliance with production requirements, and conducting stability testing. Modern analytical techniques, such as genome sequencing and flow cytometry, make it possible to accurately identify and measure probiotic strains. The result ensures that the same level of reliability and effectiveness is maintained throughout many batches [165].

Although there is increasing evidence supporting the efficacy of probiotics and prebiotics in healthcare, there is still a need for improvement in their adoption in policy and practice. Differences in regulations regarding probiotics, prebiotics, and postbiotics across nations pose challenges to global market entry and product development [166]. Regulatory harmonisation and mutual recognition agreements are essential for facilitating market entrance and upholding quality and safety standards. Establishing these agreements may assist in surmounting the existing regulatory obstacles, providing the global accessibility of secure and efficient items for customers. Effective collaboration between the business sector, academia, and regulatory agencies will be essential for promoting scientific progress and successfully navigating the regulatory environment, resulting in a significant influence on the future of healthcare.

Due to increasing consumer demand and the development of tailored medicines, market trends and prospects of probiotics, prebiotics are favourable when it comes to using these modulators to treat mental health conditions and metabolic illnesses and promote health [167]. Regulatory reforms and cooperation across the academic, biotech, and pharmaceutical industries support the development of these medicines [168]. The report “Global Probiotics Industry” estimates that the global probiotics industry will reach a value of 91.7 billion by 2030, and prebiotics anticipates a 14.9% annual growth from 2022 to 2030 (https://www.grandviewresearch.com/industry-analysis/probiotics-market, last accessed on 20 July 2024). Probiotics, prebiotics, and postbiotics have the potential to transform public health and healthcare systems by offering specific and individualised therapies for different health issues [163].

Advancements in next-generation sequencing, metagenomics, and bioinformatics have enhanced our comprehension of the intricate relationships between gut microbiota and host health, developing new therapeutics. Novel technologies, such as CRISPR-based genetic editing and high-throughput screening techniques, are used to identify and modify certain probiotic strains, enhancing their therapeutic potential [169]. AI in medicine discovery accelerates the identification of novel probiotics, prebiotics, and postbiotics, hence improving the optimisation of their compositions. Research on microencapsulation and delivery techniques enhances the stability, survivability, and targeted dispersion of probiotics, thus boosting their medicinal efficacy [170].

While probiotics, prebiotics, and postbiotics hold promising potential, there are significant challenges in comprehending their modes of action, ensuring their safety, and assessing their efficiency. These hurdles, along with the necessity for regulatory authorisation, underscore the importance of further investigation and advancement [171]. Addressing these issues and integrating these bioactive substances into conventional treatment are crucial steps. Strategic investments and collaboration among stakeholders can create an environment that supports the well-being and long-term sustainability of the global population. Using probiotics and prebiotics can help maintain the balance of beneficial gut microorganisms, strengthen the immune system, and address various health problems, including mental and gastrointestinal disorders [172]. Importantly, eubiosis not only improves patient outcomes but also reduces reliance on antibiotics and decreases healthcare expenses. Incorporating these products into preventative healthcare initiatives could significantly enhance overall well-being and alleviate the burden of chronic diseases on the healthcare system.

## 5. Conclusions

In conclusion, extensive research and clinical evidence underscore the profound impact of probiotics and prebiotics in restoring gut microbiota homeostasis and treating various diseases. It is crucial to emphasise that probiotics offer therapeutic benefits by modulating the gut microbiota, producing beneficial metabolites, and eliminating harmful substances. Similarly, prebiotics selectively stimulate beneficial microorganisms, thereby enhancing gut health and aiding in disease management. Advancements in research technologies, such as sequencing and artificial intelligence, are poised to further propel this field, addressing current challenges and unlocking new potential.

The field of microbiome-targeted nutrition and therapeutics is rapidly evolving, with novel probiotics and prebiotics emerging and expanding beyond traditional definitions. These substances are now being sourced from new, sustainable sources, reflecting the industry’s commitment to innovation and environmental responsibility. Industry trends and consumer preferences, along with advancements in delivery technologies and quality assurance, are driving the integration of these bioactive substances into various formats. While the gut remains a key focus, clinically proven applications are now extending to other body systems, including the respiratory, immune, urogenital, skin, nervous, oral, cardiometabolic, and weight-management systems; their significant role in clinical treatments will potentially revolutionise disease management and enhance overall health.

## Figures and Tables

**Figure 1 antibiotics-13-00720-f001:**
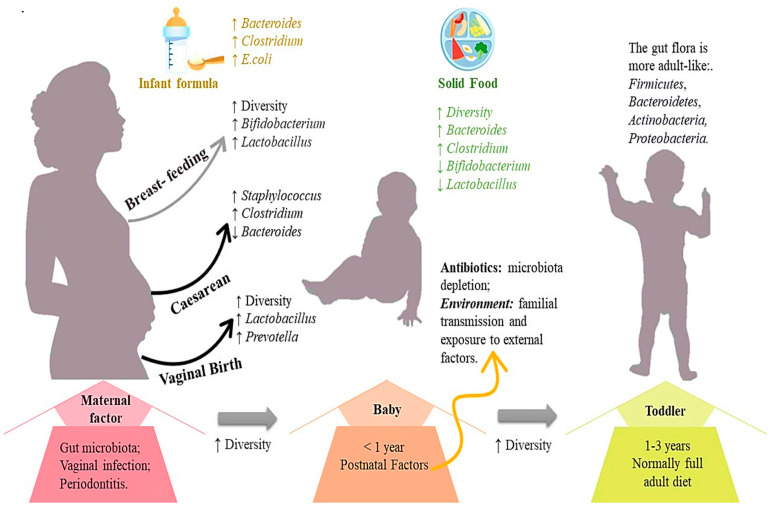
Changes in the composition of gut bacteria throughout life. The mother’s microbiota changes during pregnancy and postpartum, potentially facilitating the transmission of certain strains. Childbirth methods and food intake influence dominant bacterial groups. Over time, solid food introduces a growing bacterial variety, reaching adult levels by age three. Scheme adapted from [7,8].

**Figure 2 antibiotics-13-00720-f002:**
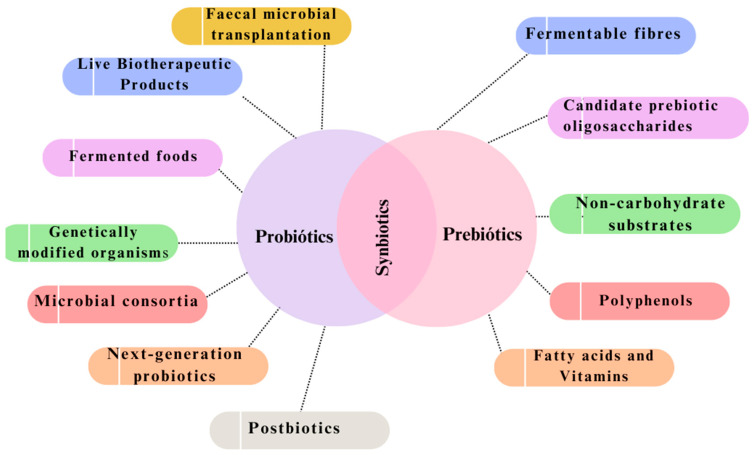
Illustration of the key concepts associated with probiotics, prebiotics, and related areas. Several emerging knowledge areas [22] intersect and connect with the established domains of probiotics and prebiotics [23].

**Figure 3 antibiotics-13-00720-f003:**
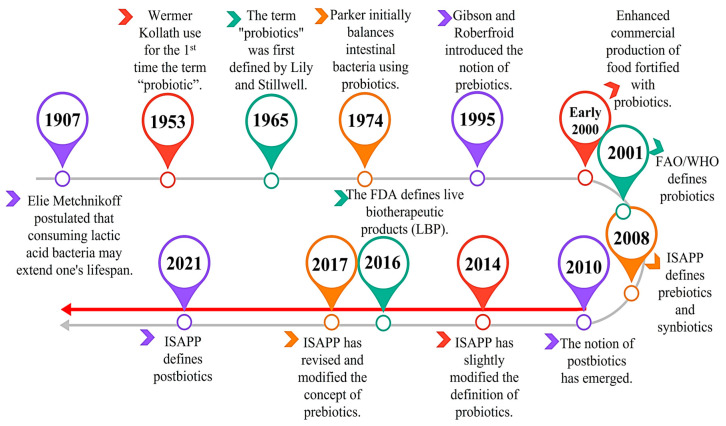
Chronological overview of the advancements in probiotics, prebiotics, synbiotics, and postbiotics as outlined by the ISAPP. Legend: the FAO refers to the Food and Agriculture Organisation of the United Nations, whereas the WHO stands for the World Health Organisation. The red arrow represents the explosion of new terms (postbiotics, paraprobiótics etc.). The information was obtained from the website https://isappscience.org/ (accesed on 11 June 2024).

**Table 1 antibiotics-13-00720-t001:** Terminology regarding the field of probiotics and prebiotics. Adapted from the ISAPP and [22].

Biogenic: refers to goods derived from or produced by living organisms, including their secretions and metabolic processes.
Probiotics: are living bacteria that provide a health advantage to the host when given in sufficient quantities.
Prebiotic: A substance that cannot be digested and metabolised by gut microbes. This process alters the makeup and activity of the gut microbiota, resulting in a positive physiological impact on the host.
Synbiotic: Refers to the combination of probiotics and prebiotics. This combination enhances the survival and establishment of living microorganisms and dietary supplements in the gastrointestinal tract. These supplements selectively promote the growth and activate the metabolism of specific beneficial bacteria, ultimately improving the well-being of the host.
Postbiotics: a bioproduct derived from inanimate microorganisms and their components that provides a health advantage to the recipient.
Live biotherapeutic product (LBP): A biological product of living organisms. It is used to prevent, treat, or cure diseases or conditions in human beings, although it is not classified as a vaccination.
Next-generation probiotic (NGP): living microorganisms found through comparative microbiota investigations that provide a health benefit when supplied in sufficient quantities.
Candidate prebiotic oligosaccharides: oligomers that satisfy current prebiotic criteria but await in vivo confirmation.
Faecal microbial transplantation: transfer of beneficial bacteria from a healthy donor into the intestines of a recipient patient using a processed liquid stool combination.
Fatty acids: carboxylic acids that have aliphatic chains and may exist in either saturated or unsaturated forms.
Fermentable fibres: refer to dietary fibres that are metabolised by microbial activity in the gastrointestinal tract.
Fermented foods: refer to food and drinks that have undergone microbial growth and activity.
Genetically modified organisms (GMOs): organisms that have undergone genetic alterations via the use of genetic engineering methods.
Microbial consortia: composed of a complex combination of microbial species that engage in symbiotic relationships, a puzzle that may consist of well-defined groups with thoroughly characterised members or unspecified combinations, inviting further exploration.
Non-carbohydrate substrates: refer to microbial growth factors that can support growth without relying on the breakdown of sugars.
Polyphenols: plant substances that naturally include phenol groups.

**Table 2 antibiotics-13-00720-t002:** Assessment of research and clinical trials that might modify the gut microbiome’s composition.

Microorganism		Target Disease and Mechanism	Ref
**Probiotics**			
**Clinical Trials**	*L. rhamnosus* TCELL-1	Assessing the Efficacy of L. Rhamnosus TCELL-1 in the Treatment of Colorectal Cancer. Phase 2 clinical trials (NCT05570942) (https://classic.clinicaltrials.gov/ct2/show/NCT05570942, last accessed on 25 June 2024)	
	BIO-25 (*Bifidobacterium breve*, *Bifidobacterium longum*, *Bifidobacterium infantis*, *Lactobacillus acidophilus*, *Lactobacillus plantarum*, *Lactobacillus paracasein*, *Lactobacillus casei*, *Bifidobacterium bifidum*, *Lactobacillus lactis*, *Lactobacillus rhamnosus*, and *Streptococcus thermophilus*)	Evaluate the therapeutic impact of the multispecies probiotic combination “BIO-25” in patients with irritable bowel syndrome (IBS) who have diarrhoea (IBS-D). Phase 4 clinical trials (NCT01667627) (https://classic.clinicaltrials.gov/ct2/show/results/NCT01667627, last accessed on 25 June 2024)	
	*Lactobacillus rhamnosus*, *Lactobacillus acidophilus*, *Lactobacillus reuteri*, *Lactobacillus paracasei*, *Lactobacillus casei*, *Lactobacillus gasseri*, *Lactobacillus plantarum*, *Bifidobacterium lactis*, *Bifidobacterium breve*, *Bifidobacterium bifidum*, *Bifidobacterium longum*, *Bifidobacterium infantis.*	Effect of Probiotic Supplementation on the Immune System in Patients with Ulcerative Colitis in Amman. Phase 3 clinical trials (NCT04223479) (https://classic.clinicaltrials.gov/ct2/show/NCT04223479, last accessed on 20 July 2024) This study aims to analyse the potential negative impacts of microorganisms on the immune system, particularly the inflammatory response. It is crucial to note that in immunocompromised individuals, the use of probiotics may carry significant risks, including the potential for infections if bacteria translocate into sterile areas, or if the introduction of probiotics disrupts the existing gut microbiota balance.	
	*Hafnia alvei* HA4597 TM	The Impact of Hafnia Alvei on Weight Loss and Glycaemic Control After Bariatric Surgery. Clinical Portuguese trials (https://classic.clinicaltrials.gov/ct2/show/NCT05170867, accessed on 25 June 2024)	
**Metabolic Diseases**	*Lactobacillus plantarum*	Decrease in body fat via breaking down lipids by oxidation.	[84]
	*Lactobacillus paracasei*, *Bifidobacterium breve*, and *Lactobacillus rhamnosus* or mixture	Decreased blood levels of lipopolysaccharide (LPS) resulted in reduced triacylglycerol concentration in the liver.	[85]
	Mixture of *Lactobacilli*, *Streptococcus*, and *Bifidobacteria*	Less weight and fat mass gain in High Fat Diet with probiotics.	[86]
	*Bacteroides thetaiotaomicron*	Reduction in total adipose tissue and rise in body mass.	[87]
	*Bifidobacterium lactis* LMG P-28149, and *Lactobacillus rhamnosus* LMG S-28148	Addressing obesity and insulin resistance by promoting the growth of *Akkermansia muciniphila* and *Rikenellaceae*. Increasing the expression of PPARγ and lipoprotein lipase. Improving the body’s response to insulin and increasing the efficiency of triglyceride elimination Reducing the abundance of *Lactobacillaceae.*	[88]
	*Bifidobacterium longum* PI10 and *Ligilactobacillus salivarius* PI2	Obesity: Increasing the expression of GLP1 and IL-10.	[89]
**Obesity**	*Lactobacillus plantarum* NK3 and *Bifidobacterium longum* NK49	Enhancing the intestinal barrier to address obesity and osteoporosis. Inhibiting the synthesis of lipopolysaccharide (LPS). Suppressing the production of TNF-α by the downregulation of NF-κB signalling.	[90]
	*Lactobacillus reuteri* LR6	Protein–energy malnutrition leads to an increase in *Bifidobacteria*, *Firmicutes*, and *Lactobacilli*.	[91]
**Diabetes**	*Akkermansia muciniphila*	Enhancement in insulin sensitivity and metabolic function leads to a reduction in inflammation and body fat accumulation.	[92,93,94]
	*Ruminococcaceae* sp. *Lachnospiraceae* sp.	It reduces weight gain and shows beneficial effects on insulin, fasting blood glucose, inflammatory markers, leptin, and chemerin levels. Moreover, dietary therapy may alter the function and composition of microbes, leading to an increase in two butyrate-producing families.	[95]
	*Lactobacillus plantarum* T3 AFB1	Decreased blood glucose levels, decreased expression of SGLT-1 and GLUT-2, decreased intestinal permeability. Probiotics engage in competitive consumption of glucose.	[96]
	*Lactobacillus fermentum* MCC2760	Enhancing the integrity of the intestinal barrier. Increasing the expression of GLUT4, GLP1, and ZO-1.	[97]
**Gut- Brain Axis**	*Lactobacillus plantarum ATCC 8014*	Reduced levels of Aβ in the hippocampus improved cognitive decline and preserved the integrity and adaptability of neurons.	[98]
	*Lactobacillus reuteri*	Autism spectrum disorders alter social and repetitive behaviours, decrease the expression of GABA receptors, and increase the expression of oxytocin in the hypothalamus.	[99]
	*Bifidobacteria*, *Lactobacillus*, *Laccoccus* and yeast. (Patent no. ZL2015104679586)	Depression can be improved by reducing depressive behaviours, decreasing neuronal cell injury, lowering Bax and cleaved caspase-3 levels, and increasing p-AKT and Bcl-2 levels through activation of the AKT signalling pathway.	[100]
	*L. brevis* DSM 27961, *L. acidophilus* DSM 32241, *L. helveticus* DSM 32242, *L. paracasei* DSM 32243, *L. plantarum* DSM 32244, *B. lactis* DSM 32246, *B. lactis* DSM 32247, *S. thermophilus* DSM 32245	Orally administering the treatment improved glucose uptake by restoring the levels of GLUT1 and GLUT3, and IGF receptor β in the brain. This was accompanied by reduced phosphorylation of AMPK and Akt. The memory of the mice also improved due to a decrease in phosphorylated tau aggregates, an increase in glycated haemoglobin, and an accumulation of advanced glycation end products.	[101]
	*Bifidobacterium longum (NK46)*	Oral dosing enhanced the gut microbiota composition, decreased blood and faecal LPS levels, elevated in the colon the quantity of tight junction proteins, and decreased the production of TNF-α and the activation of NF-κB. In addition, it decreased cognitive decline, β/γ-secretase activity, amyloid beta buildup, and caspase-3 expression in the hippocampus of mice.	[102]
	*Clostridium butyricum*	The treatment successfully reversed the GM deficit and increased the butyrate levels. It inhibited the buildup of Aβ, cognitive decline, the release of TNF-α and IL-1β, and microglia activation.	[103]
	*Lactobacillus lactis* strain carrying one plasmid (pExu)	Improved memory, lowered levels of Aβ peptides, regulated the ubiquitin–proteasome system and autophagy, and decreased neuronal inflammation and oxidative processes.	[104]
**IBS/IBD**	*Bifidobacterium longum* 35624	Irritable Bowel Syndrome (IBS). Enhanced bowel movement patterns, reduced sensitivity to internal organs, increased healing of the mucous membrane. Increased lysozyme production and increased stem niche factors. WNT3A and TGF- are two molecules.	[105]
	*Limosilactobacillus fermentum* KBL374	Inflammatory Bowel Disease (IBD). An increase in colon length, a decrease in inflammatory cytokines, an increase in body weight, and a decrease in leukocyte infiltration. Modulating immune responses, modifying gut microbiota, increasing gut barrier function.	[106]
**IBS/IBD**	*Lactobacillus johnsonii, and Lactobacillus reuteri*	Enhancing the process of tetrathionate metabolism. Reducing the presence of *Yersinia enterocolitica*.	[107,108]
	*Lactobacillus paracasei*	Colitis accompanied with a metabolic condition Manufacturing palmitoylethanolamide with the purpose of preserving intestinal functionality.	[109]
**Gut-Heart Axis**	*Lactobacillus casei*	Hypertension is associated with an increase in the abundance of *Akkermansia* and *Lactobacillus*. Reducing the ratio of *Firmicutes to Bacteroidetes* and ACE (angiotensin-converting enzyme) expression.	[39,110]
	*Bifidobacterium breve* CECT7263 and *Lactobacillus fermentum* CECT5716	Enhancing the abundance of microorganisms associated with butyrate production. Increasing the concentration of butyrate in the plasma. Minimising the formation of lipopolysaccharide (LPS).	[111]
	*Lactobacillus fermentum* CECT5716, *Lactobacillus coryniformis* CECT5711 (K8), and *Lactobacillus gasseri* CECT5714 (LC9)	Decrease NOX activity and mRNA expression of NOX-1 and NOX-4 in spontaneously hypertensive rats	[112]
	*L. reuteri* V3401	Obese persons aged 18 to 65 years with metabolic syndrome have a decreased risk of cardiovascular disease (CVD) and have lower levels of inflammatory biomarkers, including TNF-α, IL-6, IL-8, and soluble intercellular adhesion molecule-1. However, even though several studies have shown that probiotics may reduce the production of proinflammatory cytokines	[113]
**Liver**	*Bifidobacterium breve* ATCC15700	Alcoholic liver disease: Decreased endotoxemia, preserved immunological balance, reduced liver damage, increased tight junction proteins Enhance intestinal barrier function and modulate gut microbiota.	[114]
**Cancer**	*L. plantarum* YYC-3	CRC (colorectal cancer) is associated with the development of colon tumours and mucosal damage, as well as a decrease in inflammatory cytokines and the VEGF-MMP2/9 signalling pathway. Immunomodulation, changes in the composition of the gut microbiota, and the release of metabolites into the body.	[115]
	*Lactobacillus reuteri*	Melanoma Enhancing the effectiveness of immune checkpoint inhibitors (ICIs) to improve ICI response rates and patient survival. Produces indole-3-aldehyde to activate CD8+ T cells	[116]
	*Bacillus subtilis* ZK3814	Eliminates *S. aureus*, suppressed production of Agr-regulated virulence factors ZK3814	[117]
**Chronic Kidney Disease**	*Limosilactobacillus fermentum* JL-3	A 31.3% decrease in serum uric acid levels and a decrease in oxidative stress markers are characteristics of hyperuricemia. Facilitate the breakdown of uric acid and maintain the balance of gut microbes.	[118]
	*L. casei* 01	Kidney stones: Reduces the formation of renal calculi. Breaks down and makes use of oxalate.	[119]
	*L. fermentum*, *L. plantarum*, *and B. lactis*, or *L. acidophilus*, *B. bifidum*, and *B. longum*	Reduced the levels of microorganisms that cause inflammation and faecal zonulin, while increasing the levels of kynurenine in the blood.	[120]
	*Streptococcus thermophilus* KB 19, *Lactobacillus acidophilus* KB 27, and *Bifidobacterium longum* KB 31	Improvement in the standard of living. There is a decreasing trend in the levels of serum indoxyl glucuronide and C-reactive protein.	[121]
**Prebiotic**			
**Obesity**	Fermentable carbohydrate inulin	The concentration of cells that generate the hormone peptide YY (PYY) rose 87%. PYY helps decrease hunger, lower food consumption, and prevent obesity caused by diet.	[122]
	Inulin	It enhances the abundance of *Bifidobacterium* and reduces the ratio of *Firmicutes* to *Bacteroidetes*. SCFA may function as a scavenger of reactive oxygen species (ROS). In addition, it has the ability to regulate reactions to harmful bacterial attacks (LPS) and safeguard the gut from inflammatory processes. This is likely achieved by enhancing the body’s defences against reactive oxygen species (ROS) by activating colonic mucosal detoxification enzymes (GSTs). Consequently, inulin helps restore the levels of crucial proteins involved in the functioning of the intestinal smooth muscle.	[123]
	Maize starch dextrin and Lentil	Primary use in the therapy of obesity. Enhancing the abundance of *Actinobacteria* and *Bacteroidetes*. Reducing the abundance of *Firmicutes*.	[124]
	Chicory oligofructose	This therapy’s main purpose is to treat obesity. It involves increasing the levels of *Bifidobacterium* and *Collinsella*.	[125]
	Amylosucrase-modified chestnut starch	Primary use in the treatment of obesity. Enhancing the activity of the short-chain fatty acid (SCFA)–GPR43 signalling pathway.	[126]
	Fuji FF	Obesity. Increasing acetic, propionic, and butyric acid production [75].	[127]
**Obesity**	Acorn and sago polysaccharides	Primary use in the treatment of obesity and type 2 diabetes mellitus. Decreasing intestinal permeability and signs of inflammation in the mucosal lining.	[128]
	Galactooligosaccharides	Primary use in the treatment of obesity and type 2 diabetes mellitus. Enhancing the expression of GLP1. Decreasing faecal bile acid excretion.	[129]
**Diabetes**	Resistant dextrin from wheat and corn starch	The primary use of this medication is for treating type 2 diabetes mellitus. Enhancing the levels of *Akkermansia* and *Prevotella* bacteria. Increasing the activity of the IRS1-Akt-GLUT2 and SIRT1-AMPK-PPARα-CPR1α pathways.	[130]
	Isomaltodextrin	Insulin resistance. Enhancing the synthesis of acetic and butyric acids. Enhancing the integrity of the intestinal barrier. Lowering the concentration of endotoxins in the bloodstream.	[131]
**Dyslipidemia**	Whole garlic	Dyslipidemia treatment Increasing *Lachnospiraceae* decreasing *Prevotella*.	[132]
	Fucoidan and Galactooligosaccharides	Treating dyslipidemia involves enhancing the abundance of *Bacteroidetes and Proteobacteria* and promoting in *Lactobacillus casei* the activity of bile salt hydrolase. Reduce the abundance of *Actinobacteria and Firmicutes*.	[133]
	Long-chain inulin	Managing hypertension. Reducing the concentrations of acetate and propionate in faeces, as well as lowering the level of TMAO in the bloodstream.	[134]
	Glycolipids from tilapia heads	Colitis is accompanied by a metabolic disorder, such as increasing *Akkermansia*, *Allobaculum*, *Bifidobacterium*, *Coprococcus*, *Oscillospira*, *and Prevotellaceae*.	[135]
**Gut- Brain Axis**	Xylooligosaccharides	The intervention effectively reduced changes in the gut microbiota and improved cognitive function. It also reduced inflammatory responses and improved the integrity of the tight junction barrier in both the hippocampus and intestine.	[136]
	Fructooligosaccharides	The pathological changes and cognitive deficits were improved, and the levels of synapsin I and postsynaptic density protein 95, and the level of phosphorylated c-Jun N-terminal kinase decreased. In addition, the FOS administration restored the modified GM density.	[137]
**Synbiotics**			
	*Bifidobacterium* and galacto-oligosaccharides	Decreased inflammation and enhanced integrity of the intestinal barrier.	[138]
	*A. muciniphila* +inulin	Improve glucose levels.	[139]
	*Lactobacillus plantarum C29*-fermented soybean (DW2009)	The administration of DW2009 increased blood BDNF (brain-derived neurotrophic factor) levels, potentially leading to considerable improvements in cognitive and memory skills.	[137]
**Chronic Kidney Diseases**	*Lactobacillus acidophilus* and *Bifidobacterium lactis* + prebiotic (inulin)	Enhance digestive system symptoms. Decreasing trend in plasma C-reactive protein levels.	[33]
	*Lactobacillus*, *Bifidobacteria and Streptococcus genera* + prebiotic (inulin, fructooligosaccarides, and galacto-oligosaccarides	During the ongoing procedure, the main focus is on measuring the amount of indoxyl-sulfate, which is a crucial consequence. Secondary outcomes include the measurement of p-cresyl sulphate, LPS, TMAO, inflammation, and oxidative stress indicators, as well as the assessment of renal function and quality of life.	[140]
	*Lactobacillus acidophilus and Bifidobacterium lactis +* prebiotic (inulin)	CKD: Increases Bifidobacterial counts in faecal samples Reduction of Lactobacilli counts in faecal samples. Improve gastrointestinal symptoms. Slowing of progression of kidney disease.	[141]
	*Lactobacillus plantarum*, *Lactobacillus casei* subsp. *rhamnosus*, *Lactobacillus gasseri*, *Bifidobacterium infantis*, *Bifidobacterium longum*, *Lactobacillus acidophilus*, *Lactobacillus salivarius Lactobacillus sporogenes*, *and Streptococcus thermophilus* +, prebiotic (inulin and tapioca-resistant starch)	Decrease Plasma p-cresol concentration.	[142]
	*Lactobacillus casei* strain *Shirota* and *Bifidobacterium breve strain Yakult* + prebiotic (galacto-oligosaccharides)	Decrease p-Cresol. Normalisation of bowel habits. Decrease blood urea nitrogen levels.	[142]
	*Lactobacillus acidophilus* and *Bifidobacterium lactis* + prebiotic (inulin)	Enhance digestive system symptoms. Decreasing trend in plasma C-reactive protein levels.	[33]
**Obesity**	Bifidobacterium, Lactobacillus, Lactococcus, Propionibacterium plus omega-3 fatty acids	Showing a beneficial combined effect in reducing liver fat buildup and lipid accumulation compared to using probiotics alone.	[143]
	*Bacillus licheniformis plus* xylo-oligosaccharides	Indicating a beneficial combined impact on enhancing body weight increase and lipid metabolism. Reducing the abundance of *Desulfovibrionaceae and Ruminococcaceae*.	[144]
	*Lactobacillus plantarum PMO 08* plus chia seeds	Unveiling a beneficial synergistic impact on enhancing obesity. Enhancing the abundance of *Lactobacillus plantarum.*	[145]
	*Bifidobacterium lactis*, *Lactobacillus paracasei DSM 4633*, plus oat b-glucan	Elevating concentrations of acetate, propionate, and butyrate in the faeces. Reducing the amount of bile acid reservoirs.	[146]
	*Lactobacillus paracasei HII01 plus* xylo-oligosaccharides	Suppressing metabolic endotoxemia. Reducing the ratio of Firmicutes to Bacteroidetes and the presence of Enterobacteriaceae.	[147]
**NAFLD**	*Lactobacillus paracasei N1115 plus* fructo-oligosaccharides	Reducing the synthesis of lipopolysaccharides (LPS). Suppressing the expression of TLR4 and NF-κB. Boosting the functionality of the p38 MAPK pathway and elevating the levels of occludin 1 and claudin 1 expression.	[148]
	Bifidobacterium bifidum V, Lactobacillus plantarum X plus Salvia miltiorrhiza polysaccharide	Reducing liver fat accumulation and enhancing insulin sensitivity. Reducing the number of lipopolysaccharides (LPS).	[149]
	Clostridium butyricum plus corn bran	Gastrointestinal dysfunction accompanied by metabolic disease. Enhances the proliferation of bacteria that generate acetate and the synthesis of acetate and isovalerate, reducing the levels of pathogens.	[150]
**Postbiotics**			
	Peptidoglycan: *L. acidophilus*	Anti-inflammatory impact: lowering COX-2 levels and elevation of iNOS are associated with increased insulin sensitivity and glucose intolerance. Suppression of IL-12 production via the interaction of NOD2 and IRF4	[77]
	Teichoic acids: *L. plantarum L. paracasei L. rhamnosus*	Immunomodulatory action. Anti-obesogenic and anti-inflammatory effect. Inhibition of JNK, ERK, and p38 kinase phosphorylation. Improvement of phosphor-p38-AMPK levels and a reduction in NF-κB.	[151]
	Cell-Free Supernatants: *Lactobacillus* strains like *L. acidophilus, L. casei*, *L. reuteri*, *L. lactis B. longum, Saccharomyces* species like *S. boulardii*	Antioxidant activity, anti-inflammatory effect, anti-obesogenic effect, and IR reduction. Reduction in IL-6, TNF-α, and IL-1β expression. ROS and RNS scavenging properties. Scavenging free-radical DPPH. Inhibition of linoleic acid peroxidation. Decrease in TNF-α secretion and rise in IL-10 discharge. Reduction in the production of NO, COX-2, and Hsp70. Hepatic FGF21 up-regulation. FGF21–adiponectin signalling.	[152]
	Exopolysaccharides: *Bacillus* sp., *L. delbrueckii*, *L. plantarum*	Antioxidant effect, insulin resistance and type 2 diabetes regulation, anti-adipogenesis activity, hyperglycemia, and dyslipidemia improvement. Delay of atherosclerosis development inhibition of cholesterol absorption enhancement of immune response vs. pathogen ROS and RNS scavenging properties. AS160-mediated pathway. AMPK/PI3K/Akt pathway. Regulation of SCD1 (stearoyl-CoA desaturase 1), ACC (acetylCoA carboxylase), SREBP-1 (sterol regulatory element-binding protein), and FAS (fatty acid synthase). Reduction in VLDL, LDL, and triglyceride levels and increase in HDL.	[153]
	Extracellular Vesicles: *A. muciniphila, Propionibacterium freudenreichii*	The substance has anti-obesogenic and anti-inflammatory effects, reducing fat accumulation and modulating the NF-κB pathway. It also exhibits anti-inflammatory and antioxidant actions, prevents the invasion of colon cancer cells, shows antibacterial activity, promotes intestinal barrier health, and reverses impaired intestinal peristalsis induced by stress. Additionally, it prevents the invasion of entero invasive *E. coli* strains into enterocytes in vitro, improves the absorptive surface of the intestine, reduces intestinal pathogens in lambs, and aids in wound healing.	[154]
	Short chain fatty acids (butyric, propionic, and acetic acids): *Lactobacillus* spp.	To mitigate the risk of inflammatory diseases, such as obesity, diabetes type 2, or other ailments, it is important to increase energy consumption and promote the oxidation of fatty acids. This can be achieved by modulating the PGC-1α pathway through the activation of AMPK and the inhibition of HDACi (histone deacetylase inhibitors). Additionally, down-regulating PPARγ can also be beneficial. These compounds serve as an energy source, possess immunosuppressive properties, aid in energy harvesting and reduction of fat deposition, inhibit cholesterol synthesis, promote ulcerative colitis regression, block atherosclerosis, and improve insulin sensitivity, leading to a ‘statin-like effect’. Increase insulin secretion without impairing pancreatic beta cells. Metabolic disorder: Increasing *Lachnospiraceae and Proteobacteria*. Decreasing *Clostridiaceae.* NAFLD: Increasing *Blautia*, *Christensenellaceae*, and *Lactobacillus*. Increasing ZO-1 expression. Decreasing the levels of endotoxin.	[64]
	Enzymes: *Lactococcus* sp., *Lactococcus lactis, S. thermophilus, L. casei, L. fermentum, B. adolescentis*, *B. longum*, *B. infantis*, *B. breve*	This substance acts as an antioxidant and reduces inflammation in the intestines. It achieves this by scavenging free radicals through the action of catalase (CAT), glutathione peroxidase (GPx), NADH oxidase, and superoxide dismutase (SOD). These enzymes convert free radicals into oxygen (O_2_) and hydrogen peroxide (H_2_O_2_). Antioxidant activity. Possible alleviation of symptoms associated with Crohn’s disease; manipulation of gut microbiota; experimentation conducted on mice; conducted in laboratory settings; demonstrated protection against pathogens such as Giardia lamblia.	[25]
	Bacteriocins: *L. plantarum*	Anti-bacterial, anti-inflammatory, anti-obesogenic, reduce diabetes. Function on cytoplasmic membranes via pores creation. Reduction in TNF-α and IL-6 concentration. Stimulate reductions in weight gain and food intake. Decrease in PAI-1.	[155]

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
