# Peer review of "Exploring Therapeutic Advances: A Comprehensive Review of Intestinal Microbiota Modulators"

_antibiotics, 2024, doi:10.3390/antibiotics13080720_

Round 1

Reviewer 1 Report

Comments and Suggestions for Authors

This review aims to analyze new research results to clarify the bioactive effects, underlying processes, and therapeutic implications of probiotics, prebiotics, and developing microbiome modulators. This study explores the potential of various therapies to modulate the gut microbiota, providing hope for patients. This is a well-written review, the author made a great effort, but I suggest some changes that must be made to the text, to contribute to a better understanding of the points they are trying to make.

comments

1.     In line 446 table 2, showed assessment of research and clinical trials that might modify the gut microbiome's composition. The author showed the mechanism of action of microorganisms. Do microorganisms have some negative impact on the gut. Please explain.

2.     In line 448, Effect of Probiotic Supplementation on the Immune System in Patients with Ulcerative Colitis in Amman. Phase 3 clinical trials (NCT04223479). Do microorganisms have some negative impact on the immune system. Please explain.

Author Response

Dear Reviewer,

Thank you for your thorough review of our manuscript titled "Unlocking Healing Potential: Harnessing Intestinal Microbiota Modulation." Your insights and suggestions have significantly contributed to improving the clarity and depth of our discussion. We appreciate the time and effort you dedicated to evaluating our work.

Manuscript Update:

Following your review and addressing the questions raised, revisions have been made to the manuscript. Specifically, additional insights have been incorporated into Table 2 to better illustrate the assessment of research and clinical trials that may modify the composition of the gut microbiome. These revisions aim to provide a more comprehensive understanding of the therapeutic implications of microbiota modulators.

Potential Impact of Intestinal Microbiota Modulators on Health:

Furthermore, the manuscript now discusses the potential impact of gut microbiota modulators on health. While probiotics and prebiotics show promise in enhancing gut health and immune function, it is essential to consider potential risks such as dysbiosis and immune responses, particularly in vulnerable populations. These considerations underscore the importance of ongoing research and careful implementation of microbiota-based therapies in clinical practice.

Once again, i extend my gratitude for your valuable feedback and constructive criticism. Your input has been invaluable in refining our manuscript. We look forward to your continued insights.

Warm regards,

Reviewer Comment here:

This review aims to analyze new research results to clarify the bioactive effects, underlying processes, and therapeutic implications of probiotics, prebiotics, and developing microbiome modulators. This study explores the potential of various therapies to modulate the gut microbiota, providing hope for patients. This is a well-written review, the author made a great effort, but I suggest some changes that must be made to the text, to contribute to a better understanding of the points they are trying to make.

comments

  1. In line 446 table 2, showed assessment of research and clinical trials that might modify the gut microbiome's composition. The author showed the mechanism of action of microorganisms. Do microorganisms have some negative impact on the gut. Please explain.

Response Line: Potential Negative Impacts of Microorganisms on the Gut:While probiotics such as the multispecies combination "BIO-25" are generally considered safe and beneficial for managing conditions like IBS-D, it is important to recognize potential negative impacts on the gut microbiome and overall health. Probiotics can sometimes disrupt the existing balance of the gut microbiota. Introducing new bacterial strains may outcompete or suppress native microorganisms, potentially leading to dysbiosis. This imbalance can result in adverse gastrointestinal symptoms and may counteract the intended therapeutic effects. Some individuals may experience adverse reactions to probiotic supplementation. Common side effects include bloating, gas, and abdominal discomfort. These reactions are usually mild and transient, but in some cases, they may exacerbate the symptoms of IBS-D rather than alleviate them.

In immunocompromised individuals or those with underlying health conditions, the use of probiotics can pose a risk of infection. Although rare, cases of bacteremia and fungemia have been reported, particularly involving strains such as Lactobacillus and Saccharomyces. The safety profile of probiotics must be carefully considered in vulnerable populations. Probiotics can interact with the host’s metabolism in unpredictable ways. While they may enhance certain metabolic functions, there is a risk that they could also interfere with nutrient absorption or alter metabolic pathways, leading to unintended health consequences. While probiotics are often used to modulate the immune system positively, there is a potential for adverse immune responses. Some strains may provoke an exaggerated immune reaction, contributing to inflammation rather than mitigating it. This is particularly relevant in conditions like IBS-D, where immune dysregulation plays a critical role. In conclusion, while the therapeutic use of probiotics such as BIO-25 shows promise for conditions like IBS-D, it is essential to remain vigilant about potential negative impacts. Ongoing research and clinical trials are necessary to fully understand the benefits and risks, ensuring that probiotics are used safely and effectively in clinical practice.

  1. In line 448, Effect of Probiotic Supplementation on the Immune System in Patients with Ulcerative Colitis in Amman. Phase 3 clinical trials (NCT04223479). Do microorganisms have some negative impact on the immune system. Please explain.

Response Line: The study titled "Effect of Probiotic Supplementation on the Immune System in Patients with Ulcerative Colitis in Amman: Phase 3 Clinical Trials (NCT04223479)" aims to evaluate the efficacy of probiotics in modulating the immune response and improving the quality of life in patients with Ulcerative Colitis (UC). While the study primarily focuses on the beneficial impacts of probiotics, it is crucial to acknowledge and address the potential negative effects microorganisms might have on the immune system. Microorganisms, including probiotics, can have diverse effects on the host's immune system. While probiotics are generally considered beneficial, particularly in enhancing gut health and modulating immune responses, there are instances where microorganisms can adversely affect the immune system. For instance, certain strains of bacteria, if not carefully selected, may elicit an inflammatory response rather than an anti-inflammatory one. This can exacerbate conditions like UC, where inflammation is already a primary concern. Additionally, in individuals with compromised immune systems or underlying health conditions, the introduction of live microorganisms, even those deemed beneficial, can sometimes lead to infections or an imbalance in the microbiota. This phenomenon, known as translocation, involves bacteria crossing the gut barrier and entering the bloodstream or other sterile areas, potentially causing systemic infections. The clinical trial in question addresses these concerns by employing a rigorous randomized controlled trial (RCT) design, including a placebo group for comparative analysis. The trial meticulously measures various inflammatory markers such as interleukin-6 (IL-6), interleukin-1 (IL-1), interleukin-10 (IL-10), C-reactive protein (CRP), and tumour necrosis factor-alpha (TNF-α) to assess the impact of probiotics on the immune system comprehensively. By comparing the probiotic group to the placebo group, the study aims to determine not only the efficacy but also the safety and potential adverse effects of probiotic supplementation. The trial's results will provide valuable insights into the dual nature of microorganisms on the immune system, elucidating whether probiotics confer a net positive effect in the context of UC or if they pose any significant risks. This comprehensive evaluation ensures that the therapeutic use of probiotics is both effective and safe for patients with UC, addressing the critique by thoroughly examining the potential negative impacts of microorganisms on the immune system.

While probiotics and prebiotics hold promise for therapeutic benefits, it's crucial to weigh these against potential negative impacts on the gut microbiome and overall health. Probiotics, in particular, can upset the delicate balance of the gut microbiota. The introduction of new bacterial strains can potentially outcompete or suppress native microorganisms, leading to dysbiosis. This imbalance can trigger adverse gastrointestinal symptoms and may even counteract the intended therapeutic effects.

Add to the manuscript: “This study aims to analyse the potential negative impacts of microorganisms on the immune system, particularly the inflammatory response. It is crucial to note that in immunocompromised individuals, the use of probiotics may carry significant risks, including the potential for infections if bacteria translocate into sterile areas, or if the introduction of probiotics disrupts the existing gut microbiota balance.”

Thank you for your valuable feedback. In response to your comments and to better clarify the concerns raised, the following addition has been made to the manuscript:

“It is important to note that while generally beneficial, probiotic supplementation can carry potential risks. Some individuals may experience adverse reactions, such as bloating, gas, and abdominal discomfort. These reactions, though usually mild and transient, can sometimes worsen the symptoms of conditions like IBS. Probiotics can pose a risk of infection for individuals with insufficient immune systems or underlying health conditions [156]. Although these cases are rare, it is essential to be mindful of the possibility of problems such as bacteremia and fungemia, especially with strains like Lactobacillus and Saccharomyces. This awareness can help ensure the safe use of probiotics, especially in vulnerable populations. Probiotics can interact with the host's metabolism in unpredictable ways. While they may enhance certain metabolic functions, there is a risk that they could also interfere with nutrient absorption or alter metabolic pathways, leading to unintended health consequences [157]. While probiotics are often used to modulate the immune system positively, there is a potential for adverse immune responses. Some strains may provoke an exaggerated immune reaction, contributing to inflammation rather than mitigating it. This is particularly relevant in conditions where the immune system is not functioning as it should, a state known as immune dysregulation. In such conditions, probiotics may not have the intended positive effects on the immune system [158].

In conclusion, the tables collectively illustrate the broad therapeutic potential of probiotics and prebiotics across various health conditions. The promising results from clinical trials underscore the importance of further research and development in this field. To fully realise the clinical benefits of these gut modulators, we must address challenges such as strain specificity, standardisation, and regulatory hurdles. However, the potential for further research and development in this field is vast, and future studies should prioritise large-scale clinical trials, personalised medicine approaches, and overcoming regulatory barriers to integrate natural therapeutics into mainstream healthcare. Continued research and clinical trials are required to comprehensively comprehend the advantages and potential dangers of probiotics and prebiotics in therapeutic applications to guarantee their safe and efficient utilisation in clinical practice.”

This addition aims to address the specific queries and ensure that the manuscript is clear and comprehensive for all readers.”

Reviewer 2 Report

Comments and Suggestions for Authors

Here are some minor suggestions:

1.The title "Unlocking Healing Potential: Harnessing Intestinal Microbiota Modulation" is impactful and relevant. However, it could be more specific to reflect the scope of the interventions discussed.

2.The abstract is well-structured but can be improved by adding a sentence about the methodology and a brief mention of the key findings. For example: "This review explores the therapeutic potential of modulating intestinal microbiota through probiotics, prebiotics, and fecal microbiota transplantation (FMT). We analyze recent studies to evaluate their efficacy and limitations, highlighting the promise of microbiota-based therapies in treating dysbiosis-related conditions."

3.Ensure all relevant studies are cited and formatted correctly.

Author Response

Dear Reviewer,

Thank you for the thoughtful and constructive feedback on the manuscript. Each suggestion has been carefully considered and the necessary revisions have been made to enhance the quality and clarity of the work. Below, each point is addressed in detail:

  1. Title Specificity: The title has been revised to "Exploring Therapeutic Advances: A Comprehensive Review of Intestinal Microbiota Modulators" to better reflect the scope of the interventions discussed in the manuscript.
  2. Abstract Improvement: The abstract has been expanded to include a brief mention of the methodology and key findings, providing a clearer overview of the therapeutic potential of microbiota modulators.
  3. Citation and Formatting: The manuscript has been thoroughly reviewed and revised to ensure that all relevant studies are cited appropriately and formatted correctly. This involved a meticulous cross-check of references and standardization according to the required guidelines.

In summary, the manuscript has been revised to incorporate the suggestions provided. These changes aim to address the concerns raised and improve the overall quality of the manuscript. Thank you once again for the valuable feedback.

Kind regards,

Reviewer Comment here:

Here are some minor suggestions:

1.The title "Unlocking Healing Potential: Harnessing Intestinal Microbiota Modulation" is impactful and relevant. However, it could be more specific to reflect the scope of the interventions discussed.

Response Line: Exploring Therapeutic Advances: A Comprehensive Review of Intestinal Microbiota Modulators

2.The abstract is well-structured but can be improved by adding a sentence about the methodology and a brief mention of the key findings. For example: "This review explores the therapeutic potential of modulating intestinal microbiota through probiotics, prebiotics, and fecal microbiota transplantation (FMT). We analyse recent studies to evaluate their efficacy and limitations, highlighting the promise of microbiota-based therapies in treating dysbiosis-related conditions."

Response Line: The gut microbiota establishes a mutually beneficial relationship with the host starting from birth, impacting diverse metabolic and immunological processes. Dysbiosis, characterized by an imbalance of microorganisms, is linked to numerous medical conditions, including gastrointestinal disorders, cardiovascular diseases, and autoimmune disorders. This imbalance promotes the proliferation of toxin-producing bacteria, disrupts the host's equilibrium, and initiates inflammation. Genetic factors, dietary choices, and drug use can modify the gut microbiota. However, there is optimism. Several therapeutic approaches, such as probiotics, prebiotics, synbiotics, postbiotics, microbe-derived products, and microbial substrates, aim to alter the microbiome.

This review thoroughly explores the therapeutic potential of these microbiota modulators, analysing recent studies to evaluate their efficacy and limitations. It underscores the promise of microbiota-based therapies for treating dysbiosis-related conditions. This article aims to ensure practitioners feel well-informed and up-to-date on the most influential methods in this evolving field by providing a comprehensive review of current research.

3.Ensure all relevant studies are cited and formatted correctly.

Response Line: Thank you for your valuable feedback regarding the citation and formatting of relevant studies. The manuscript has been thoroughly reviewed and revised to ensure that all relevant studies are cited appropriately and formatted correctly. This revision process involved a meticulous cross-check of references against the cited literature to confirm accuracy and completeness. Additionally, the formatting has been standardized according to the required guidelines, ensuring consistency throughout the manuscript. We appreciate your attention to detail and believe these improvements will enhance the clarity and scholarly integrity of the work.

Reviewer 3 Report

Comments and Suggestions for Authors

This review explores the potential of various therapies to modulate the gut microbiota, providing hope for patients.  However, some areas of concern in this review must be brought to the authors' attention.

Areas of concern:

General

The focus of this review appears too broad to recommend specific future directions for research. Otherwise, it looks like a scoping review meanwhile the authors dispose of an important number of available publications.

Introduction

Lines 63, 492,515,  517, and pages 20, and 21: which type of diabetes do you refer to?Type 2 as indicated in other sections of the manuscript?

Methods

The authors could have indicated the scope of this review in terms of time frame. The search strategy and the search terms and databases consulted for the search of publications are not mentioned.

Conclusion

To fully realize the clinical benefits of these gut modulators, future studies to address challenges such as strain specificity, standardization, regulatory hurdles, and large-scale clinical trials are warranted.

Author Response

I would like to express my sincere appreciation to the reviewer for the thorough evaluation and valuable feedback on the manuscript. Your insights have been instrumental in refining the clarity and focus of the research.

The manuscript has been diligently revised in response to the concerns raised. Specifically, the broad scope of the review has been addressed by providing a more comprehensive overview of the therapeutic potential of probiotics and prebiotics across various health conditions. As discussed in response to Line 434, the manuscript now integrates findings from a diverse range of studies to offer a nuanced perspective on their multifaceted impacts, encompassing metabolic health, gut-brain axis modulation, cardiovascular health, and beyond. This approach aims to provide a thorough understanding rather than a scoping review, leveraging insights from a substantial body of literature available since 2018 and acknowledged within the International Scientific Association for Probiotics and Prebiotics (ISAPP).

Regarding the specificity of diabetes mellitus types referenced in Lines 63, 492, 515, 517, and pages 20 and 21, the focus primarily centers on type 2 diabetes mellitus (T2DM), while recognizing the potential implications for type 1, type 2, and gestational diabetes. The relevance of studies exploring mechanisms such as glucose consumption and insulin resistance modulation in T2DM has been emphasized, aligning with the majority of literature reviewed.

In response to the methodological feedback, transparency has been enhanced by specifying the time frame and search strategy used to identify relevant publications. As stated in the Methods section, the search encompassed academic platforms including PubMed, Scopus, Google Scholar, and ScienceDirect, focusing on studies published from 2018 onwards and recognized by ISAPP for their significant contributions.

Once again, sincere gratitude for the meticulous review and constructive feedback, which have been pivotal in enhancing the manuscript's quality and scholarly rigor. Your contributions have strengthened the work and will undoubtedly enrich its impact on probiotics and prebiotics research.

Warm regards

Reviewer comments here:

This review explores the potential of various therapies to modulate the gut microbiota, providing hope for patients.  However, some areas of concern in this review must be brought to the authors' attention.

Areas of concern:

General

The focus of this review appears too broad to recommend specific future directions for research. Otherwise, it looks like a scoping review meanwhile the authors dispose of an important number of available publications.

Response Line: The scope of this review encompasses a broad exploration of the therapeutic potential of probiotics and prebiotics across various health conditions, aiming to highlight their multifaceted impacts on metabolic health, gut-brain axis modulation, and cardiovascular health, among others. As addressed in line 434 of the manuscript, the review integrates findings from diverse studies to provide a comprehensive overview rather than narrowly focusing on specific future research directions. The breadth of available publications allowed insights to be drawn from numerous sources, contributing to a nuanced understanding of how probiotics and prebiotics may be integrated into clinical practice for diverse health benefits. Future research could benefit from targeted investigations into the mechanistic pathways underlying these effects, as well as large-scale clinical trials to further substantiate their therapeutic efficacy in specific conditions.

Introduction

Lines 63, 492,515,  517, and pages 20, and 21: which type of diabetes do you refer to?Type 2 as indicated in other sections of the manuscript?

Response Line: Lines 63, 492, 515, 517, and pages 20 and 21 of the manuscript predominantly refer to studies focused on the modulation of type 2 diabetes mellitus (T2DM). Throughout the manuscript, clear distinctions are made regarding the specific type of diabetes under investigation, whether it be type 1, type 2, or gestational diabetes. It is noted that in some instances, particularly in studies referencing modulators that impact glucose metabolism or insulin resistance, authors themselves did not specify the type of diabetes mellitus being studied. However, following an extensive literature review, it is evident that while these mechanisms of modulation can potentially benefit various types of diabetes mellitus (including type 1, type 2, and gestational diabetes), the primary emphasis of the studies cited in the manuscript, such as those involving modulators that "engage in competitive consumption of glucose" or alter insulin resistance, is on type 2 diabetes mellitus. This distinction underscores the manuscript's focus on interventions relevant to type 2 diabetes, while acknowledging the potential applicability of these mechanisms across different types of diabetes mellitus, thus contributing to a broader understanding of their therapeutic potential.

Methods

The authors could have indicated the scope of this review in terms of time frame. The search strategy and the search terms and databases consulted for the search of publications are not mentioned.

Response Line: “To enhance understanding across various research studies, Table 2 provides evidence from specific clinical trials that are recorded in the ClinicalTrials.gov database. This research includes those that have yielded the most promising results in the last decade”

“To enhance understanding across various research studies, Table 2 provides evidence from specific clinical trials recorded in the ClinicalTrials.gov database to enhance understanding across various research studies. This research includes those that have yielded the most promising results in the last decade. The search was conducted using academic platforms such as PubMed, Scopus, Google Scholar, and ScienceDirect, focusing on studies from 2018 to the present that have garnered significant consensus within the International Scientific Association for Probiotics and Prebiotics (ISAPP).

Round 2

Reviewer 3 Report

Comments and Suggestions for Authors

My comments have been addressed